# Outcomes of Patients Undergoing Closed Traction Coronary Endarterectomy: A Long-Term Single Center Study [note 1]

**DOI:** 10.3390/jcm11237026

**Published:** 2022-11-28

**Authors:** Sharaf-Eldin Shehada, Fanar Mourad, Ali Haddad, Belal Darwish, Noura Ryadi, Ilir Balaj, Heinz Jakob, Arjang Ruhparwar

**Affiliations:** 1West German Heart and Vascular Centre, Department of Thoracic and Cardiovascular Surgery, University Hospital Essen, University Duisburg-Essen, Hufelandstraße 55, 45122 Essen, Germany; 2Department of Anaesthesiology and Intensive Care Medicine, University Hospital Essen, University Duisburg-Essen, Hufelandstraße 55, 45147 Essen, Germany

**Keywords:** coronary endarterectomy, severe and diffuse coronary artery disease, high SYNTAX score

## Abstract

Background—Coronary endarterectomy (CEA) is an option for treating severely diffused coronary artery diseases; however, many surgeons avoid performing it due to its complexity and reported controversial results. Therefore, we aimed to review the results of patients undergoing CEA within coronary artery bypass grafting (CABG). Methods—This is a retrospective observational study evaluating the results of patients undergoing CEA within CABG surgery between March 2003 and February 2018. Follow-up via active personal and/or telephone interviews was performed to evaluate long-term clinical outcomes. The study endpoints included early postoperative incidence of myocardial infarction or cardiac mortality, long-term survival, and freedom from major adverse cardiac and cerebrovascular events (MACCE). Results—A total of 326 patients were included in this study for evaluation. The patients’ mean age was 67 years; 88% were male, and most presented with three-vessel disease, reporting a mean SYNTAX score of 33.1 ± 12. Approximately 5.5% (*n* = 18) of the patients had undergone previous CABG surgery. A total of 394 CEAs within a mean of 4.3 ± 1.1 grafts per patient were performed. The indication for CEA was either totally (*n* = 111, 28.2%) or sub-totally (*n* = 283, 71.8%) occluded coronary arteries. Early results included perioperative myocardial infarction in eight (2.4%), stroke in eight (2.4%), and in-hospital mortality in thirteen (4.0%) patients. Long-term clinical follow-up reported mortality in 27.6% and overall incidence of MACCE in 41.4% of the patients at the ten-year follow-up. Conclusions—Patients with severe and diffuse CAD are difficult candidates for surgical revascularization. CEA offers an option to allow complete revascularization, even in the case of chronic occlusion, when the myocardium is still viable. The closed traction CEA technique presented here is our preferred method; it achieves satisfactory short- and long-term results.

## 1. Introduction

Coronary endarterectomy (CEA) was introduced as an option to treat severely diffused coronary artery disease (CAD) [1], but surgeons prefer not to perform CEA due to its reported conflicting results [2,3]. CEA can be performed via traction (closed) or direct vision (open) techniques. In the direct vision technique, a longitudinal incision is made across the diseased coronary artery, and the atheromatous plaque is carefully removed from the wall of the main vessel and side branches. Then, the vessel is reconstructed with a patched saphenous vein combined with left internal mammary artery (LIMA) grafting [4] or direct grafting using the LIMA only [5]. The traction technique is performed with one direct incision of the diseased coronary artery, after which the atheromatous plaque is carefully dissected from the lamina of the vascular wall with gentle traction applied to remove the atheromatous cylinder from the main vessel. A secondary incision can be made to remove the atheroma from the side branches; accordingly, one or more grafts are added to the endarterectomized coronary arteries [6]. We modified this closed traction technique and are presenting our results in this study. Moreover, we evaluated patients’ early and late clinical outcomes.

## 2. Methods

### 2.1. Study Design

This is a retrospective observational study. From March 2003 to February 2018, a total of 425 patients presenting with severely diffused coronary artery disease undergoing endarterectomy during CABG surgery were evaluated. Ninety-nine patients who underwent concomitant surgery were excluded from the study, which resulted in a study cohort of 326 patients. Preoperative, operative, and postoperative patient data were prospectively collected in our institutional database and retrospectively evaluated. All available preoperative angiographies were individually examined, and the extent of coronary artery disease was classified according to SYNTAX scoring by a board-certified cardiologist. Postoperative coronary angiography was evaluated when available. Long-term clinical follow-up was conducted via an active personal interview using a standardized questionnaire, which was developed based on the EuroQol questionnaire [7], by two cardiac surgeons. Review board approval from the ethical committee (Ref# 18-8420-BO) was obtained before study initiation.

### 2.2. Indication and Surgical Technique

After initiating cardiopulmonary bypass and cross clamping the aorta [8], coronary endarterectomy was performed in cases of total or sub-total occlusion (i.e., a 1.25 mm probe could not be gently introduced through the heavily atherosclerotic narrowed lumen to the vessel’s distal portion). The criterion for performing CEA was the vessel’s outer diameter and viable surrounding myocardium representing a significant amount of myocardial tissue worth preservation. The applied closed traction technique is a standard technique in our institution and consists of the following five steps: (1) Single incision of the coronary artery at an area deemed to be appropriate for anastomosis. (2) Gentle traction of the obstructing distal atheromatous cylinder. Ideally, a long and smoothly tapered cylinder is extracted (Figure 1A,C). (3) Proximal traction with intended disruption of the atheromatous cylinder to avoid massive competitive flow. (4) In the case of distal disruption of the atheromatous cylinder during traction, a second distal incision (Figure 1B) with subsequent additional CEA is undertaken if judged appropriate. Alternatively, the distal part of the proximally endarterectomized vessel can be left untouched if there is adequate distal run-off. (5) Flushing of the coronary vessel using cardioplegia and simultaneous massage of the proximal and distal vessel areas to remove any potential residual atheromatous debris. Thereafter, distal end-to-side anastomosis is undertaken with the chosen graft, followed by sequential side-to-side anastomosis at the primary incision site. An illustrating step-by-step case report on a patient undergoing CABG + CEA in all coronary territories was recently published showing this modified closed traction technique [9]. After bypass grafting, transit time flow measurement (TTFM; MediStim, Oslo, Norway) [10] was applied in all cases before and after CPB weaning to ensure adequate graft function in all grafts.

### 2.3. Postoperative Treatment

The postoperative management regimen for patients undergoing CABG + CEA surgery consists of heparinization within 4 h, followed by intravenous aspirin within 8 h, followed by oral aspirin from the first postoperative day. Thereafter, patients are advised to take life-long aspirin. Since 2008, application of a P2Y_12_ inhibitor within 24 h and for 6 months was added to the regimen to prevent early platelet aggregation and fast endothelization after CEA. “Multiplate^®^’’ analyser (Roche Diagnostics, Mannheim, Germany) [11,12] testing is used to adjust the dosage efficacy of the given P2Y_12_ inhibitor.

### 2.4. Study Endpoints and Definitions

The study endpoints included early incidence of graft failure, myocardial infarction, and in-hospital mortality, in addition to long-term survival and freedom from major adverse cardiac and cerebrovascular events (MACCE). Nonelective surgery was defined as having an urgent or emergent indication, as described by the Society of Thoracic Surgeons database. Low cardiac output syndrome was defined as perioperative need for intra-aortic balloon pump (IABP) or extracorporeal membrane oxygenator (ECMO) support. Postoperative myocardial infarction (MI) was defined according to the Third Universal Definition of MI [13]. A cerebrovascular event was defined as the occurrence of a new stroke. Major adverse cerebrovascular and cardiac events included incidence of overall mortality, myocardial infarction, stroke, or need for reintervention (stenting or re-CABG), as defined in the SYNTAX trial [14].

### 2.5. Statistics

All statistical analyses were performed using SPSS software (version 22.0, IBM Corp., Armonk, NY, USA). Continuous data were expressed as the mean ± standard deviation (SD) or median and interquartile (25th and 75th) ranges (IQR). Categorical data were expressed as frequencies and percentages. Kaplan–Meier curves were generated to estimate the long-term survival function and freedom from major adverse cardiac and cerebrovascular events.

## 3. Results

### 3.1. Preoperative Data

A total of 326 consecutive patients over a 15-year period who presented with severe and diffuse coronary artery disease (mean SYNTAX score of 33.1 ± 12) and underwent isolated CABG + CEA were included for evaluation. The mean age was 66.7 ± 9.3 years, and most patients were male (287, 88%). Three-vessel disease was reported in most (302, 93%) patients, and 123 (37.7%) patients had experienced a previous myocardial infarction. Eighteen patients (5.5%) had previously undergone CABG. Approximately one-third (99, 30.4%) of the patients presented for urgent or emergent surgery, and 92 (28.2%) patients had highly impaired left ventricular function (ejection fraction < 30%). Detailed baseline characteristics are summarized in Table 1.

### 3.2. Postoperative Outcomes

Table 2 summarizes the intraoperative outcomes. All operations were performed under cardiopulmonary bypass with a mean cross-clamp time of 84.3 ± 19.2 min. A total of 394 CEAs within a mean of 4.3 ± 1.1 grafts per patient were constructed, as 57 (17.5%) patients required more than one CEA. The indication for CEA was either totally (111, 28.2%) or sub-totally (283, 71.8%) occluded coronary arteries involving the LAD territory (174, 44.2%), the RCA territory (153, 38.8%), and/or the LCX-territory (67, 17%). The grafts used after CEA were either arterial in 145 (36.8%) cases or venous in 249 (63.2%) cases. The transit time blood flow measurement after discontinuation of ECC resulted in 67.5 ± 39.7 (mL/min mean) over the CEA grafts. Postoperatively, the patients received either single (152, 46.6%) or dual antiplatelet therapy (APT) (174, 53.4%). Early postoperative outcomes are reported in Table 3, which shows an incidence of stroke in eight (2.4%) patients, perioperative myocardial infarction in eight (2.4%) patients, and in-hospital mortality in thirteen (4%) patients.

### 3.3. Long-Term Outcomes

Follow-up imaging using either conventional coronary angiography (Figure 2) or multi-slice coronary computed tomography angiography was available in one-quarter of the patients. These imaging results were recently published, with a graft patency rate of 89.9% for non-CEA grafts versus 84.5% for CEA grafts [8]. Clinical follow-up for at least one year, with a median of 8.9 (IQR: 3–12) years, was complete for 321 patients (98.5%) by March 2020. Table 4 summarizes detailed long-term outcomes, which include ten-year mortality of 90 (27.6%) patients and an overall incidence of MACCE in 135 (41.4%) patients (Figure 3A,B). In Figure 4, the clinical status of the surviving patients is classified according to their NYHA class: 132 patients presented in NYHA class I, 61 in NYHA class II, 14 in NYHA class III, and 2 in NYHA class IV. Additionally, 11 patients developed strokes, and 15 patients required percutaneous revascularization during follow-up. Ten of those were on the basis of myocardial infarction, where five were related to CEA grafts presenting after 1, 3, 98, 108, and 190 months, and the other five had non-CEA graft infarction. No re-CABG surgery took place, but five patients underwent another cardiac surgery. Ten patients required pacemaker implantation. Finally, a significant difference between patients receiving dual- versus single-APT was observed in regard to overall long-term survival (79.3% vs. 50%, *p* < 0.001), incidence of myocardial infarction (2.2% vs. 9.6%, *p* = 0.006), percutaneous revascularization (4.4% vs. 12.3%, *p* = 0.018), and MACCE (27% vs. 57.9%, *p* < 0.001).

## 4. Discussion

Treatment of patients with severe and diffuse coronary artery disease is challenging; these patients are often not candidates for percutaneous coronary intervention or even surgical myocardial revascularization. Consequently, coronary endarterectomy has been introduced as an option to allow revascularization in these patients [1]. However, it is not routinely performed due to its complexity and reported controversial results, and only experienced surgeons performing this procedure [15,16].

Outcomes after coronary endarterectomy are a matter of debate, with reported results varying from acceptable to poor. According to a meta-analysis by Soylu et al., CEA had significantly higher 30-day mortality (odds ratios (OR) = 1.69, *p* < 0.00001) and perioperative (OR = 2.10, *p* < 0.00001) and postoperative MI (OR = 3.34, *p* = 0.0003) when the results were compared with CABG alone [4]. Similarly, another meta-analysis by Wang et al. reported that CEA + CABG was associated with significantly higher 30-day mortality compared with isolated CABG (OR = 1.86, *p* < 0.0001) [3]. In another study, Myers et al. reported an operative mortality of 3–4.1% and an incidence of perioperative myocardial infarction of 4–4.1%, depending on the CEA technique, either using a vein patch or direct LAD grafting, respectively [17]. More recently, the group of Nishigawa in Tokyo reported a 30-day mortality rate of 1.1% and a perioperative myocardial infarction rate of 9% after CEA of the LAD and LITA grafting [18].

Currently, two CEA techniques can be applied: the traction (closed) technique [15] or the direct vision (open) technique [19,20]. The benefits, limitations, and outcomes of both techniques have been widely discussed, with controversial results [3,4,5,18,19,20]. In this study, we present the outcomes of a modified closed traction CEA technique, which we considered suitable to enable revascularization when surrounding myocardial areas were viable and deemed worthy of being preserved. Different investigations can be used to define viable myocardium; cardiac magnetic resonance imaging (CMR), single-photon emission CT (SPECT), and positron emission tomography (PET) imaging with F18-fluorodeoxyglucose (FDG) are the most commonly used modalities for assessing myocardial viability [21]. In the late 1990s, dobutamine stress echocardiography became one of the most acceptable myocardium vitality tests [22,23]. It was adopted increasingly over time and became a routine and first-line noninvasive diagnostic tool to evaluate myocardial viability. CMR was used in the current study to test myocardial viability in patients presenting with severely impaired ventricular function; otherwise, preoperative and/or intraoperative dobutamine stress echocardiography, which are routine tests, were used to define and diagnose myocardial function and viability in patients with less severe ventricular impairment. In most (72%) cases, the coronary vessels were sub-totally occluded with a caliber smaller than 1.25 mm and poor run-off; in 28% of cases, the coronary vessels were totally occluded. This modified technique is based on a combination of steps for successful revascularization: a single incision followed by gentle extraction of the atheromatous cylinder distally ending smoothly and tapered, and intended disruption of the cylinder proximally to avoid competitive flow. Coronary vessel flushing with cardioplegia and vessel massage is used to eliminate the debris that follows. Then, grafting is performed, followed by TTFM to test its function immediately. In the rare case of significantly reduced flow into the CEA area, immediate revision is indicated. In the current study, in-hospital mortality was reported in 13 (4%) patients. The cause of death of 11 patients was cardiac-related. Moreover, one patient developed intestinal ischemia and multi-organ failure, and another patient developed refractory multi-organ failure on top of sepsis. Eight (2.4%) patients had perioperative myocardial infarction, which is associated with early graft failure. Another eight patients developed strokes, which could mainly be attributed to severe global atherosclerosis and calcifications affecting the ascending aorta and supra-aortic vessels.

Our group recently reported imaging results after CABG surgery concomitant with closed traction CEA [8]. Unfortunately, follow-up imaging was only available for 85 (26%) patients, mainly due to financial issues, although we proposed it for all patients. To compensate for this limitation, clinical follow-up through a personal and/or telephone call interview using a standardized questionnaire, closely related to the European Quality of Life questionnaire [7], was conducted, with a 98.5% completion rate. The questionnaire was administered from December 2019 to March 2020, thus ensuring a minimum follow-up time of 12 months, with a median of 8.9 years. Overall survival was 64.1% (209 patients), and freedom from MACCE was 57.1% (186 patients). This is comparable to a recently published nationwide Danish population study of patients undergoing CABG surgery only [24]. The clinical status of the surviving patients is illustrated in Figure 4, showing that 132 patients presented in NYHA class I, 61 in NYHA class II, 14 in NYHA class III, and 2 in NYHA class IV. Among the surviving patients, 15 underwent percutaneous revascularization; 10 were due to MI (due to significant stenosis or occlusion of the CEA graft in 5 patients and non-CEA graft occlusion in the other 5 patients). The remaining five patients underwent reintervention for progression of their coronary artery disease.

It is known that graft patency is affected by different factors, including previous myocardial infarction in the graft region, poor run-off distally to the endarterectomized vessel, and the type of graft used, as well as intraoperative transient time flow measurement on the endarterectomized graft. In our recently reported series, graft flow in the patent vessels was 67.4 ± 38, in contrast to occluded grafts (46.1 ± 37) (*p* = 0.05), at a mean imaging follow-up time of 53 ± 49 months [8]. It should be noted that most of the occluded grafts were venous (14 out of 16). Similar results were reported in other studies that compared arterial and venous graft patency [25,26]. An underestimated factor for graft patency might be the role of early dual antiplatelet therapy starting on the first postoperative day using aspirin + P2Y_12_ inhibitor for 6 months, which was introduced into our postoperative strategy in 2008, in contrast to single antiplatelet therapy with aspirin alone throughout the whole study period. In fact, at their last follow-up, patients with dual-APT showed significantly better survival (79.3% vs. 50%, *p* < 0.001) and lower incidence of myocardial infarction (2.2% vs. 9.6%, *p* = 0.006) and percutaneous revascularization (4.4% vs. 12.3%, *p* = 0.018) compared to those treated with aspirin alone. Our current hypothesis suggests that the use of dual antiplatelet therapy could protect the injured remnants of the vessel’s media against fast neointimal tissue ingrowth, which is in accordance with ESC/EACTS and ACC/AHA guidelines recommending dual antiplatelet therapy for at least 6 months after drug-eluting stent implantation [27,28].

## 5. Study Limitations

This study has numerous limitations based on its nature and characteristics. One limitation of the study is the lack of a control group treated conservatively with optimal medical therapy (OMT), percutaneous coronary artery intervention (PCI), or CABG surgery without coronary endarterectomy. We could not include such a control group as CEA is a bail-out strategy, with the decision to perform it only made intraoperatively. Another limitation is that this is a retrospective observational study in which the determination to perform CEA could not be predicted, making prospective randomization impossible for the procedure. Finally, postoperative imaging, the most important control tool, was only available in 26% of the patients since most were asymptomatic during the follow-up period and thus refused to undergo elective imaging given the cost of imaging would not be covered based on the reimbursement strategy of the health system.

## 6. Conclusions

Our modified closed traction coronary endarterectomy technique offers an option to achieve complete surgical revascularization in patients with severely diffused coronary artery disease with satisfactory short- and long-term outcomes. When supported by concomitant dual antiplatelet therapy, long-term outcomes can probably be further improved. Thus, the armamentarium for surgical revascularization of end-stage coronary artery disease can be supplemented with this bail-out procedure. A controlled randomized trial should be undertaken in the future.

## Figures and Tables

**Figure 1 jcm-11-07026-f001:**
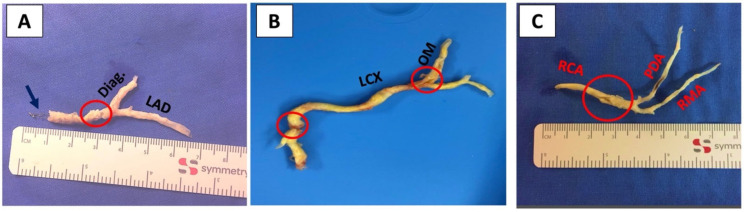
Various atheromatous cylinders extracted from different coronary territories. Legend: The red circle represents the site of incision(s) from (**A**) the LAD territory and its diagonal, where the black arrow indicates a stent that was extracted from within the atheroma; (**B**) the LCX territory and its marginal branch, where the cylinder was disrupted and a second incision was needed; and (**C**) the RCA territory. Diag = Diagonal branch, LAD = Left anterior descending artery, LCX = Left circumflex artery, OM = Obtuse marginal, RCA = Right coronary artery, PDA = Posterior descending artery, RMA = Right marginal artery.

**Figure 2 jcm-11-07026-f002:**
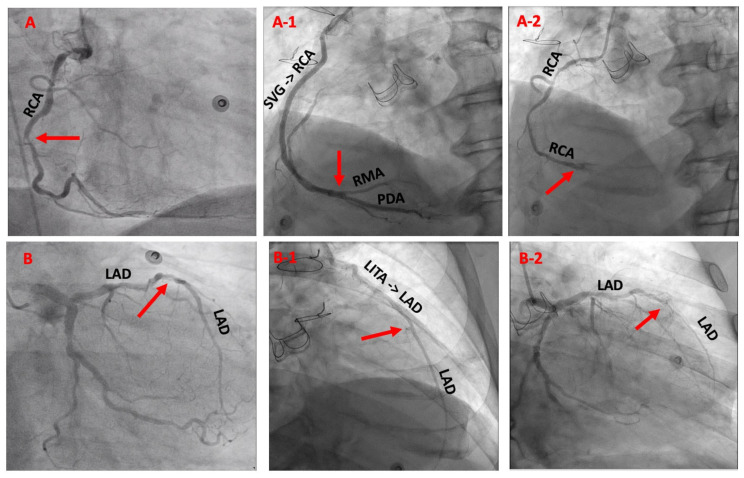
Preoperative and six-year postoperative coronary angiography status post-aortic valve replacement and CABG with CEA for subtotally occluded RCA and LAD. Indication for control angiography was prosthetic valve endocarditis prior to aortic prosthesis replacement. Legends: (**A**) preoperative RCA: (**A-1**) shows patent vein graft to the RCA with adequate run-off, (**A-2**) shows patent native RCA after CEA; (**B**) preoperative LAD: (**B-1**) shows patent LITA graft to the LAD with smooth distal vessel contour, (**B-2**) shows patent native LAD after CEA. The red arrows indicate the preoperative site of occlusion and postoperative site of anastomosis. RCA = Right coronary artery, SVG->RCA = Saphenous vein grafting to the right coronary artery, LAD = Left anterior descending artery, LITA->LAD = Left internal thoracic artery bypass to the left anterior descending artery.

**Figure 3 jcm-11-07026-f003:**
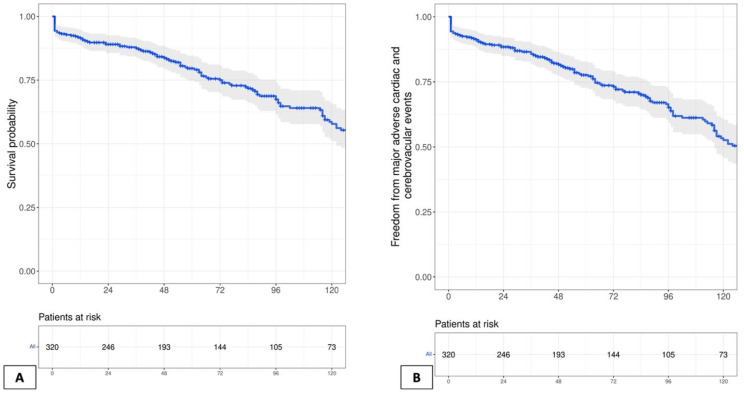
Long-term clinical outcomes presented as: (**A**) Kaplan–Meier curve showing estimated cumulative survival, (**B**) Kaplan–Meier curve showing freedom from MACCE.

**Figure 4 jcm-11-07026-f004:**
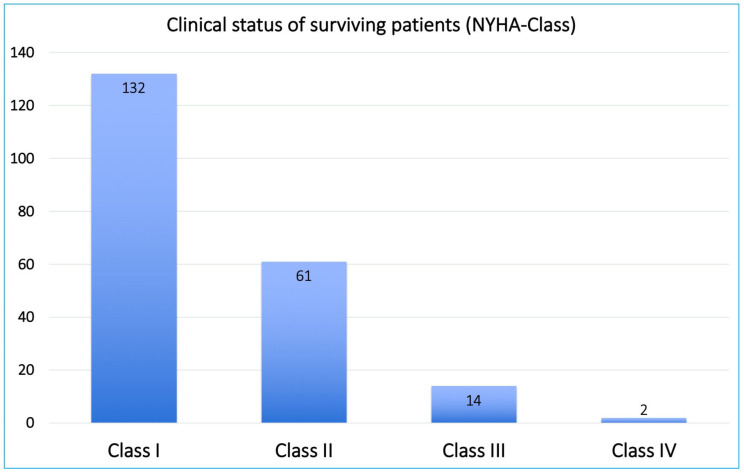
Chart bars showing the clinical status of surviving patients based on their NYHA class.

**Table 1 jcm-11-07026-t001:** Baseline characteristics.

	All Patients (*n* = 326)
Age, years	66.7 ± 9.3
Gender, males	287 (88)
Body mass index, kg/m^2^	27.4 ± 4.1

Diabetes mellitus	125 (38.3)
Nicotine abuse	115 (35.3)
Hypercholesterinemia	204 (62.6)
Chronic obstructive pulmonary diseases	38 (11.6)
Peripheral vascular disease	53 (16.3)
Preoperative dialysis	6 (1.8)

Previous cerebrovascular event	25 (7.7)
Previous myocardial infarction	123 (37.7)
Previous CABG	18 (5.5)
NYHA class III–IV	108 (33.1)
Nonelective surgery	99 (30.4)

Extent of coronary artery disease	
Three-vessel disease	302 (93)
Two-vessel disease	21 (6)
One-vessel disease	3 (1)
SYNTAX score	33.1 ± 12

Left ventricular function	
EF > 50%	143 (43.9)
EF = 35–50%	91 (27.9)
EF < 30%	92 (28.2)

Data are presented as mean ± SD or number and (%). NYHA = New York Heart Association, PTCA = percutaneous transluminal coronary angioplasty, CABG = coronary artery bypass grafting, EF = ejection fraction.

**Table 2 jcm-11-07026-t002:** Intraoperative outcomes.

	All Patients (*n* = 326)
Aortic cross-clamp time	84.3 ± 19.2
Number of grafts for each patient	4.3 ± 1.1
Total number of CEA grafts	394
One CEA	269 (82.5)
Two CEAs	47 (14.4)
Three CEAs	9 (2.8)
Four CEAs	1 (0.3)
Indication for CEA	
Totally occluded vessel	111 (28.2)
Sub-totally occluded vessel	283 (71.8)
CEA vessel	
LAD territory	174 (44.2)
RCA territory	153 (38.8)
LCX territory	67 (17)
Graft used after CEA	
Arterial graft	145 (36.8)
Venous graft	249 (63.2)
TTFM after CEA, mL/min	67.5 ± 39.7

Data are presented as mean ± SD or number and (%). CEA = coronary endarterectomy, LAD = left anterior descending artery, LCX = left circumflex artery, RCA = right coronary artery, TTFM = transit time flow measurement.

**Table 3 jcm-11-07026-t003:** Postoperative outcomes.

	All Patients (*n* = 326)
Low cardiac output syndrome	
Need for IABP	20 (6.1)
Need for ECMO	5 (1.5)
Myocardial infarction	8 (2.4)
In-hospital mortality	13 (4)
Cardiac-related mortality	11 (3.3)

Cerebrovascular events	
Transient ischemic attack	3 (0.9)
Stroke	8 (2.4)

Re-exploration for bleeding	18 (5.5)
ICU stay, days	2.4 ± 2.1
Postoperative dialysis	22 (6.7)
Respiratory complications	
Need for reintubation	23 (7)
Need for tracheostomy	22 (6.7)

Antiplatelet therapy (APT)	
Single APT	152 (46.6)
Dual APT	174 (53.4)

Data are presented as mean ± SD or number and (%). IAPB = intra-aortic balloon pump, ECMO = extracorporeal membrane oxygenation, ICU = intensive care unit.

**Table 4 jcm-11-07026-t004:** Long-term clinical outcomes.

Follow-Up Results	All Patients 326 (100)	Single-APT (*n* = 152)	Dual-APT (*n* = 174)	*p*-Value
Lost during follow-up	5 (1.5)	3 (2)	2 (1.1)	
One-year mortality	27 (8.3)	15 (9.8)	12 (6.9)	0.32
Ten-year mortality	90 (27.6)	56 (36.8)	34 (19.5)	<0.001
Overall mortality at last follow-up	112 (34.4)	76 (50)	36 (20.7)	<0.001
Overall MACCE at last follow-up	135 (41.4)	88 (57.9)	47 (27)	<0.001
Follow-up, median (IQR) years	8.9 (3–12)			

**Survivors’ clinical follow-up**	**209 (100)**	**(*n* = 73)**	**(*n* = 136)**	
NYHA class I–II	193 (92.3)	59 (80.8)	129 (94.9)	0.018
NYHA class III–IV	16 (7.6)	9 (12.3)	7 (5.1)	0.018
Stroke	11 (5.2)	5 (6.8)	6 (4.4)	0.25
Myocardial infarction	10 (4.8)	7 (9.6)	3 (2.2)	0.006
PCI/stenting	15 (7.1)	9 (12.3)	6 (4.4)	0.018
Re-CABG	0			
Other cardiac surgery	5 (2.4)			
Pacemaker implantation	10 (4.8)			

Data are presented as number and (%). MACCE = major adverse cardiac and cerebrovascular events, IQR = interquartile range, NYHA = New York Heart Association functional classification, PCI = percutaneous coronary intervention, CABG = coronary artery bypass grafting.

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
