# Peer review of "Outcomes of Patients Undergoing Closed Traction Coronary Endarterectomy: A Long-Term Single Center Study†"

_jcm, 2022, doi:10.3390/jcm11237026_

Round 1
Reviewer 1 Report
The reviewer congratulates the authors for performing a retrospective observational study on an important subject, evaluating the results of patients undergoing CEA within CABG surgery. They concluded that modified closed-traction coronary endarterectomy technique offers an option to enable complete surgical revascularization in patients with severe and diffuse coronary artery disease with satisfactory short and long-term outcomes. This involves great efforts, however, there are some issues that need to be addressed.
1. The criteria for performing CEA was the vessel ́s outer diameter and viable surrounding myocardium representing a significant amount of myocardial tissue worth to be preserved. How to define viable surrounding myocardium representing a significant amount of myocardial tissue?
2. With regard to postoperative treatment, what does “multiplate” testing refer to?
3. The clinical outcomes revealed significant results between arterial graft and venous graft used after CEA. What is the criteria in using arterial graft after CEA?
4. A significant difference between patients receiving dual- versus single-APT was observed in regard to overall long-term survival (79.3% vs. 50%, p<0.001), incidence of myocardial infarction (2.2% vs. 9.6%, p=0.006), percutaneous revascularization (4.4% vs. 175 12.3%, p=0.018), and MACCE (27% vs. 57.9%, p<0.001). Is there any multivariate analysis performed? More importantly, the authors compared the antiplatelet strategy. However, the clinical outcomes improved as the advances of the overall treatment strategy. This should be taken into consideration when comparing the antiplatelet therapy.
Author Response
The reviewer congratulates the authors for performing a retrospective observational study on an important subject, evaluating the results of patients undergoing CEA within CABG surgery. They concluded that modified closed-traction coronary endarterectomy technique offers an option to enable complete surgical revascularization in patients with severe and diffuse coronary artery disease with satisfactory short and long-term outcomes. This involves great efforts, however, there are some issues that need to be addressed.
Reply: We thank the reviewer for this comment. We replied the reviewer’s raised concerns and issues as follow:
- The criteria for performing CEA was the vessel ́s outer diameter and viable surrounding myocardium representing a significant amount of myocardial tissue worth to be preserved. How to define viable surrounding myocardium representing a significant amount of myocardial tissue?
Reply 1. We thank the reviewer for this important comment. Basically, different tests could be used to define viable myocardium, usually the use of Cardiac Magnetic Resonance imaging (CMR), Single-Photon Emission CT (SPECT), Cardiac Magnetic Resonance Imaging (CMR), and Positron Emission Tomography (PET) imagining with F18-fluorodeoxyglocse (FDG) are the most common used modalities for assessing myocardial viability [19]. Since the late nineties of the last century dobutamine stress echocardiography became one of the most acceptable myocardium vitality tests [20, 21]. Based on this, dobutamine stress echocardiography gained more adoption over time and became a routine and first line noninvasive diagnostic tool to define myocardial viability. Similarly, in the current study CMR was used to test myocardial viability in patients presented with severely impaired ventricular functions, otherwise preoperative and/ or intraoperative dobutamine stress echocardiography which is a routine testing was used to define and diagnose myocardial functions and viability in those patients. This comment has been added to the discussion to avoid any confusion.
Changes 1: See page 8-9, Lines 226-238. Citations number 21-23
***************************************************************************
- With regard to postoperative treatment, what does “multiplateâ” testing refer to?
Reply 2. We thank the reviewer for this comment. Multiplateâ testing has been used to investigate platelet functions under the effect aspirin and P2Y12 inhibitors. We added the citation of Multiplateâ investigators explaining the testing and management of this testing.
Changes 2: See page 3, Lines 102-104. Citations number 11-12
***************************************************************************
- The clinical outcomes revealed significant results between arterial graft and venous graft used after CEA. What is the criteria in using arterial graft after CEA?
Reply 3. We thank the reviewer for this comment. Actually, we do not have a special criteria to use any of the grafts after CEA. As most of other institutions, LIMA has been the favorite graft used to revascularize the LAD territory. Right internal mammary and radial arteries were used when patients were younger irrelevant of their coronary vessels status and if CEA was performed or not. As the reviewers know, the mean age of patients in the current study were around 67 years. Additionally, grafts were prepared pre-revascularization and CEA was performed as described as a bail out strategy when other options were not available.
Changes 3: None
***************************************************************************
- A significant difference between patients receiving dual- versus single-APT was observed in regard to overall long-term survival (79.3% vs. 50%, p<0.001), incidence of myocardial infarction (2.2% vs. 9.6%, p=0.006), percutaneous revascularization (4.4% vs. 175 12.3%, p=0.018), and MACCE (27% vs. 57.9%, p<0.001). Is there any multivariate analysis performed? More importantly, the authors compared the antiplatelet strategy. However, the clinical outcomes improved as the advances of the overall treatment strategy. This should be taken into consideration when comparing the antiplatelet therapy.
Reply 4. We thank the reviewer for this comment. We totally agree with the reviewer in this regard. That’s why we actually are now examining the difference between both treatment strategies in those patients in a widely basis and we are preparing an article in this regard, which supposed to address this point in details and see if there are other factors affecting long-term results etc.
Changes 4: Please check page 9-10, lines 281-288. Please check the marked changes in the whole manuscript.

Reviewer 2 Report
The study is formally correct, well written and with a good number of patients but takes into consideration a technique that over the years has given very controversial results. Most likely the long term results of this technique are very influenced by the technical skills of the surgeon and the center in which it operates. The main message that emerges is the importance of using dual antiplatelet therapy, as already indicated by the international guidelines. Therefore the message launched by this work does not add new information to what is already known in the literature and the results are probably not fully reproducible on a large scale.
Author Response
Reply: We thank the reviewer for this insightful comment and we totally agree with point that the results are surgeon and center related as any of the other cardiac surgery procedures. To be noted that in the current study we have presented this CEA technique is a bail out technique and not as routine technique for all surgeons. Of course as many of the complex cardiac surgeries, CEA can be performed from an experienced surgeons and the early and long-term results would be affected accordingly. However, we disagree with the reviewer regarding the point that the message of this work does not add new information, as the reviewer knows many surgeons avoid this procedure due to its controversial results and we wanted to share our experience with this technique and support the theory of doing this procedure as a last chance option when required especially that the number of patients presenting with severely diseased and end-stage coronary arteries is increasing steadily, especially with the increasing of PCI and stenting technologies. Even though some changes have been made to the introduction, citations and conclusion as recommended from the reviewer aiming to further improve the quality of this article.
Changes: Please check the marked changes in the whole manuscript.

Round 2
Reviewer 2 Report
The changes made by the authors have certainly improved the manuscript as a whole, allowing the reader to better appreciate the message to consider this technique as a rescue in case of complex patients.
Author Response
Reply: We thank the reviewer for this comment.
Changes: Some changes have been done for the limitation section.